# Study of Township Construction Land Carrying Capacity and Spatial Pattern Matching in Loess Plateau Hilly and Gully Region: A Case of Xifeng in China

**DOI:** 10.3390/ijerph192316316

**Published:** 2022-12-06

**Authors:** Yirui Zhao, Tongsheng Li, Julin Li, Mengwei Song

**Affiliations:** 1College of Urban and Environmental Sciences, Northwest University, Xi’an 710127, China; 2Shaanxi Provincial Research Institute, Xi’an 710127, China

**Keywords:** townships, construction land carrying capacity, spatial pattern matching, Xifeng

## Abstract

With the acceleration of urbanization, the construction land scale of urban and rural areas is constantly expanding, which leads to contradiction and conflict between territorial development and ecological protection becoming more and more serious. Therefore, as an important unit of county (district), and even urban and rural, development, the study on land resource carrying capacity and the rationality of the development can provide some basis for developing the optimal strategies of differential territorial space. Taking Xifeng, Gansu Province, China as the research area, this study constructs the evaluation index system of township construction land carrying capacity from the three dimensions of ecological protection, natural environment, and social economy. It evaluates the suitability of township construction land by the means of a comprehensive scoring method and discusses the carrying capacity and spatial pattern matching of township construction land based on the suitability evaluation results. The results showed that: (1) the spatial difference of suitability of construction land is obvious, which is higher in the city center than in the surrounding areas; (2) the comprehensive carrying capacity of township construction land is 52.62%, and different townships range from 3.78% to 13.15%. It is different between towns; (3) on the whole, the condition of township construction land is well-developed, and the main distribution forms are flaky, banded, and dotted. (4) There is a positive correlation between spatial matching and carrying capacity. The carrying capacity should be considered in regional development to avoid overdevelopment. It can provide a basis for optimizing the territorial spatial layout, strengthening the coordinated development among townships, and improving the comprehensive township carrying capacity in the Loess Plateau hilly and gully region.

## 1. Introduction

Land is the all-important resource and material basis for human survival and development [1], which is related to food security, social security, and eco-security. Land resource carrying capacity is a significant component of resource and environment carrying capacity. From 1950 to 2018, with the rapid growth of the global urban population and the urbanization rate, the demand of land resources for urban development increased on a regular basis, which has inevitably contributed to disordered expansion and the deterioration of the ecological environment [2]. Land shortage and ecological destruction have become the restricting factors affecting global sustainable development. At present, every country has achieved considerable economic and social development, which results in the decline of the quality of the ecological environment and land carrying capacity [3]. So as to mitigate the negative impacts of urbanization, the 2012 “Rio+20” summit launched the “Earth of the Future” research plan as an essential strategic research agenda, which combines the research on resource carrying capacity and the research on global change and sustainable development closely [4]. China also attaches great importance to the carrying capacity and sustainability of resources and the environment. In 2015, the central committee of the communist party of China and the state council issued the Opinions on Accelerating the Construction of Ecological Civilization, which emphasized that we should strictly observe the ecological red line of resources and environment and limit all kinds of development activities beyond the carrying capacity of resources and environment. The Loess Plateau hilly and gully region is one of the most vulnerable regions nationwide in China, as well as worldwide. The regional ecosystem is seriously damaged, due to the interference of natural and human factors, thereby aggravating the man–land conflicts and seriously affecting the development and utilization of territorial space and regional sustainable development [5,6]. With the purpose of ensuring the rational and orderly development of land resources, the central government uses spatial planning to strictly control the scale and guide the space of the land. In this context, it is imperative to carry out relevant research on land resources to play the leading role of decision making management and to realize the intensive and economical utilization of urban and rural land resources and the sustainable development of regional society and the economy.

Land resource carrying capacity, as one of the crucial bases to measure whether the human and land relationship is coordinated, has been concerned for a long time. According to Manning, the concept of carrying capacity was first mentioned in 1936, which belongs to the category of physical research [7]. Since then, it has been widely applied in population, economy, ecology, and other disciplines [8]. Especially since 1940, global problems, such as population expansion, resource scarcity, and deterioration of the ecological environment, have become increasingly prominent, and countries have carried out studies on carrying capacity relationships in accordance with human beings and land resources [9]. The influential research is the research on land resource carrying capacity in Australia in 1973 [10]. By the beginning of 1980, the Food and Agriculture Organization of the United Nations (FAO) has conducted the study on the population carrying capacity of land, which proves to be beneficial to promoting global and regional economic and social planning and sustainable development [11,12]. It was not until 1986 that a study on the productive capacity of land resources and population carrying capacity in China was conducted by the Committee of Comprehensive Investigation of Natural Resources, under the Chinese Academy of Sciences, which set a precedent for the study of land carrying capacity in China [13]. After that, numerous scholars carried out a series of assessments on natural resources and environmental carrying capacity, including land, water, energy, etc. The development of cities and towns has triggered a series of problems, such as the reduction of ecological space, the deterioration of the environment, the confusion of spatial layout, etc., which is considered to be related to the land carrying capacity [14,15]. The new round of spatial planning in China is also aimed at solving the issues such as excessive occupation of ecological space, ecological damage, and environmental pollution caused by disordered, excessive, and decentralized development through constraining resources and environment carrying capacity. Therefore, the space development management and control system, based on the resources and environment carrying capacity, has become a significant path to achieve regional coordinated economic and social and environmental development [16,17].

Regarding the concept of land carrying capacity, UNESCO defines it as “the maintenance of an acceptable level of subsistence in an area that can withstand the intensity of human activities” [18]. On the grounds of relevant research achievements at home and abroad, this paper intends to summarize from three dimensions: research content, research methods, and research scale. However, in foreign countries, carrying capacity strength analysis is mainly carried out based on human activities [19], social economy [20], land use [21,22], and supply and demand equilibrium [23], as well as the influencing factors [24,25], resource and environment carrying capacity prediction [26], etc. Methods such as the ETCC model [27], spatial analysis methods of geographic information systems (GIS) and scoring methods [28], the descriptive method [29], and spatial dynamics model [30] are mainly adopted. The scale mainly involves watershed [31,32], provinces [33], and counties [34]. In China, the content primarily aims at the definition of concept connotation [35], the analysis of regional dynamic evolution [36,37,38], and the diagnosis of influencing factors [39,40]; the methods mainly include the evaluation of ecological sensitivity [41], the projection pursuit model [42,43], the improved ecological footprint model [44,45], the fuzzy comprehensive evaluation method [46], and the TOPSIS model [47,48]; the scale principally includes large and medium-sized urban agglomeration [49,50], central cities [51], and typical functional area [52]. In summary, contents have shifted from a single element-oriented evaluation to a multi-objective and multi-angle comprehensive research. Methods have become more diversified and comprehensive, and they have strengthened scientification and applicability, compared with the past. The evaluation scale showed a multi-scale evolution trend. However, the scholars currently have not yet reached an agreement on the concept of land carrying capacity, and the theoretical framework and indicator system are not perfect. In addition, the research focuses more on the comparative analysis and prediction of the spatial and temporal characteristics of land carrying capacity, and it is relatively weak to explore the internal spatial differentiation of micro scale units in different typical regions.

The hilly and gully region of Loess Plateau is a significant ecological reservation in the middle reaches of the Yellow River and also is a key area for China to consolidate poverty alleviation [53]. With the continuous progress of urbanization and industrialization, the limited land resources in the hilly and gully regions of the Loess Plateau are facing huge challenges. The land carrying capacity has become a key factor restricting the vitality and sustainable development of the economic development in the hilly and gully regions of the Loess Plateau. Therefore, this paper selects Xifeng as an empirical area, takes townships as the research object, uses comprehensive scores, spatial analysis, and qualitative description methods to analyze the spatial differentiation characteristics of land carrying capacity among townships in Xifeng, and puts forward targeted policy recommendations, with a view to providing a practical basis for the intensive and economical use of land resources in rural areas of hilly and gully regions of the Loess Plateau and spatial optimization of towns and townships in the new era.

## 2. Materials and Methods

### 2.1. The Study Area

Xifeng is located in the east of Gansu Province in China, which belongs to Qingyang. The geographical location is between 107°27′ E–107°52′ E, 35°25′ N–35°5′ N (Figure 1). It is an essential part of the “The First Plateau in the world”. It forms unique morphological characteristics with vertical and horizontal gullies because of loose soil on the surface of the loess layer, undulating beams, and ridges and erosion from rain and rivers. It also has serious soil erosion, which belongs to typical the Loess Plateau hilly and gully area. The district, with a total area of 999.45 km^2^, has 3 sub-district offices, 7 townships, and 100 administrative villages. In 2020, the Seventh National Census Data showed that the resident population in the region was 513,500. In 2019, the Third National Land Survey Data showed that total area of construction land in the region was 105.68 km^2^, which accounted for 10.68% of the total area in the region. The per capita construction land area was 205.80 m^2^/person, which was significantly larger than townships per capita construction land standard.

### 2.2. Data Sources

The basic data related to the study includes DEM, geological disasters, land use, permanent basic farmland, and ecological protection redline in Xifeng. Among them, the 30 m resolution digital elevation model (DEM) data came from the geospatial data cloud website (http://www.stats.gov.cn/ (accessed on 22 March 2022)); the geological disaster data were derived from the geological disaster prevention and control planning of Xifeng (2011–2020) and the 2020 investigation report on potential geological disasters. The vector data of land use, permanent basic farmland, and ecological reserves are all from the updated database at the unified time point of the third national land survey of Xifeng District Land and Resources Bureau in 2019. According to the land-use regulation system of the Law of Land Administration of the People’s Republic of China, this paper specifically involves land-use types, including current construction land, general cultivated land, forest land, grassland, garden plot, wetland, unused land, and others.

Through processing basic data by ArcGIS10.4, current land can be obtained, and permanent basic farmland, 30 m × 30 m grid cells of ecological protection redline area and multi-ring buffer results of national highways, provincial roads, and county roads were obtained. Then, the suitability evaluation grade map was formed by using superposition of spatial weighted and natural break point method.

### 2.3. Research Framework

With the continuous advancement of urbanization, sustainable land use has become the key to the sustainable development in different regions. In order to solve the increasingly intensified spatial imbalance in the process of territorial spatial reconstruction and to coordinate the regional social, economic, and ecological relations, it is necessary to carry out the research on the carrying capacity of construction land and the matching degree of spatial pattern between present utilization status and rational development of land use. The specific research framework: ① the selection and processing of suitability evaluation indicators, ② the generation of township construction land suitability evaluation grade map, ③ calculating the carrying capacity and development potential of townships construction land, and ④ spatial pattern matching degree evaluation and correlation analysis. The research process is shown in Figure 2.

### 2.4. The Township Construction Land Suitability Evaluation

#### 2.4.1. Indicator System Construction

Whether the land can be developed into construction land is constrained by various factors. The paper takes the “double evaluation” technical guideline as the fundamental basis, based on relevant researches [40,54]. A total of 10 single elements were selected from the three aspects involving the ecological protection, natural environment, and social economy, constructing an evaluation index system combining with reality of research area, according to the degree of the constraint of each factor on land development and utilization, which were divided into the strongly binding indicators and the less strongly binding indicators.

For the construction of the indicator system, the hilly and gully region of Loess Plateau is a typical ecologically fragile area in China [55]. According to the principle of ecological priority, the ecological protection red line is regarded as a strong binding indicator [56]. The ecological red line is an important space to maintain the ecological security pattern and ensure the integrity of the ecological service system, which is a key element in coordinating the relationship between ecological environment and land use [57,58]. The rainfall is concentrated and intense in the Loess Plateau hilly and gully region, thereby resulting in a severe soil erosion [59], a decline in soil fertility, and a serious reduction in grain production [60,61]. So as to ensure regional grain self-sufficiency and prevent arable land from becoming non-agricultural and non-graining, the permanent basic farmland was taken as a strong binding indicator. The utilization way and irrational structure of water resources in the Loess Plateau has triggered problems such as river interruption, wetlands shrinkage, and water pollution in local areas. Drought and water shortage prove to be a considerable obstacle to the high-quality development of the Loess Plateau hilly and gully region [62,63]. The water area is also used as a strong binding indicator [64]. The unique geomorphological morphology of the Loess Plateau hilly and gully region contributes to the poor slope stability and great surface elevation fluctuation. Land construction in areas with large slope or large elevation fluctuation could give rise to ecological damage with ease [65,66]. Therefore, the slope and elevation are crucial indicators to measure the topography of the loess region. In recent years, rainfall erosion in the hilly and gully region of Loess Plateau has been intense [67]. Under the action of geological movement, geological disasters, such as landslides, plateau collapse, and ground subsidence, have occurred frequently, which poses a threat to the lives and property safety of residents and increases the difficulty of engineering construction [68,69]. In addition, the current land use limits the development of future construction to a certain extent. The distance between construction land and roads (national highway, provincial roads, and county roads) could reflect the convenience of residents’ travel, affect the accessibility of information flow and logistics, and enhance the land value. Therefore, the above single elements should be considered to be applied to construct an index system and carry out the evaluation of the suitability of townships construction land.

On the basis of relevant technical reference documents and previous research [70,71,72,73], for the strongly binding indicators, the Boolean value method was used to assign 0 and 1 values to the indicators; for the less strongly binding indicators, and the Delphi method was used to assign 0–10 value, according to the degree of constraint. The weight values of each factor were determined by analytic hierarchy and expert scoring (Table 1).

#### 2.4.2. Township Construction Land Suitability Evaluation

The suitability evaluation of construction land is the requirement of land quality for planning and construction within a certain geographical scope, that is, the economic and feasibility evaluation of urban and rural development and construction based on the comprehensive influence of multiple factors, such as ecology, transportation, location, economy, and policies [74]. Through the comprehensive score method, the comprehensive value of each township construction land suitability can be calculated. The calculation formula is as follows [75]:(1)$$E=\prod_{i=1}^{m}F_{i}\times \sum_{j=1}^{n}w_{j}f_{j}$$

In Formula (1), *E* is the comprehensive suitability evaluation score; *i* is the serial number of the strongly binding indicators; *j* is the serial number of the less strongly binding indicators; *F_i_* represents the *i*th strongly binding indicators suitability evaluation score; *f_j_* is the *j*th less strongly binding indicators suitability evaluation score; *w_j_* stands for the *j*th less strongly binding indicators weight; *m* is the total number of strongly binding indicators; *n* is the total number of the less strongly binding indicators.

### 2.5. Township Construction Land Carrying Capacity Evaluation

Based on the suitability evaluation results of township construction, from the perspective of production and living suitability, the most suitable, the more suitable, the generally suitable, the less suitable, and the unsuitable areas were regarded as the most suitable carrying, the more suitable carrying, the general carrying, the less suitable carrying, and unsuitable carrying area. Among them, the most suitable, the more suitable, and the generally suitable areas were collectively regarded as the suitable construction area. The current construction land is set as the development area. The suitable construction area and the development region were regarded as the “critical carrying area”, which reflects the maximum scale and intensity of human social and economic activities that the construction land can carry. The overlapping area between the suitable construction area and the developed area is defined as the developed suitable construction area. Based on previous studies [76], the formula of the critical carrying capacity of construction land is:(2)$$P_{1}=\frac{(C_{1}+C_{2}+C_{3})\cup C_{s}}{U}\times 100\%$$

In Formula (2), *P*_1_ represents the critical value of the carrying capacity of construction land; *C*_1_, *C*_2_, and *C*_3_ represent the area of the most suitable carrying area, the more suitable carrying, and the generally suitable carrying area; *C_S_* indicates the total area of current construction land; *U* indicates the total area of townships in Xifeng.

Formula application: take Dong Zhi as an example, when *C*_1_ = 16.74, *C*_2_ = 26.17, *C*_3_ = 31.1, *C_S_* = 28.08, and *U* = 989.51, then *P*_1_ = 10.32%.

### 2.6. Township Construction Land Development Potential Evaluation

If the critical value of the construction land carrying capacity is regarded as the ultimate development intensity, then the ratio of the current construction land area and the total area of the township are regarded as the developed intensity, and the difference between the ultimate development intensity and the developed intensity is the remaining development intensity, which is the development potential of the township construction land. The calculation formula is:(3)$$P_{2}=1-\frac{C}{(C_{1}+C_{2}+C_{3})\cup C_{S}}\times 100\%$$

In Formula (3), *P*_2_ is the remaining development intensity of construction land, that is, development potential; *C* is the area of the developed suitable area.

As far as Formulas (2) and (3) are concerned, the larger the *P*_1_, the stronger the carrying capacity of the township construction land, and the greater the *P*_2_, the greater the remaining development potential of the township construction land.

Formula application: take Dong Zhi as an example, when *C* = 28.08, then *P*_2_ = 7.48%.

### 2.7. Spatial Pattern Matching Evaluation

The spatial pattern matching is a variable that reflects the degree of spatial coupling between the current construction land and the loadable interval of construction land [77]. ArcGIS10.4 is used to superimpose the suitable construction area and the developed area of the township, and then the carrying ability of the developed land of the township is determined. Following that, the spatial pattern matching degree of the current construction land and the bearable area of the township construction land could be calculated. Calculated as follows:(4)$$M=\frac{\left(C_{1}+C_{2}+C_{3}\right)}{C_{S}}\times 100\%$$

In Formula (4), *C*_1_, *C*_2_, and *C*_3_ represent the overlapping area of the current construction land, the suitable carrying area, the more suitable carrying area, and the general suitable carrying area, respectively; *C_S_* indicates the total area of the current construction land in the township.

Formula application: take Dong Zhi as an example, *M* = 85.47%.

## 3. The Analysis of Results

### 3.1. Suitability Evaluation Analysis of Township Construction Land

The suitability evaluation results of township in the study area were obtained by ArcGIS 10.4 processing (Table 2 and Figure 3). From the evaluation results, it could be seen that the most suitable area is 58.15 km^2^, the more suitable area is 100.47 km^2^, the generally suitable area is 140.33 km^2^, the less suitable area is 215.87 km^2^, and the unsuitable area is 474.69 km^2^ of township construction land in Xifeng District. Among them, the unsuitable area accounts for the largest proportion about 47.97% of total area of the townships; the suitable area has the smallest proportion, only 5.88% of total area of the townships; the proportion of the more suitable areas, the generally suitable areas, and the less suitable areas are, in order, 10.15%, 14.18%, and 21.82%.

From Figure 3, there are great differences in the spatial suitability and agglomeration degree of township construction land in Xifeng. The most suitable areas are relatively concentrated and flaky. They are mainly distributed in Dongzhi, Wenquan, Houguanzhai, and Pengyuan Town because of some villages in these townships are distributed near urban areas and located within urban development boundaries. Furthermore, the geographical location and the terrain are so superior and convenient, and the flat loess surfaces are less constrained by ecological environmental protection. In addition, they are suitable for development and utilization as construction land. The more suitable are concentrated and contiguously distributed in the peripheral of the most suitable areas, which is obvious in Dongzhi and Xiaojin, and mainly rely on the economic radiation of city and townships and convenient external traffic conditions, in which case they can be used as construction land for development. However, the ecological protection and actual situation of townships should be given priority to avoid soil erosion and subsidence of the plateau. The generally suitable areas are mostly distributed in the areas with large topographic relief and constrained by the ecological environment of the Loess Plateau hilly and gully region, which are mostly belts and are dominant in every township. Due to the constraints of terrain conditions, traffic conditions, ecological protection, and other conditions, the development space is limited, and the difficulty is increased, so the suitability degree is general. Less suitable and unsuitable areas are widely distributed in each township, accounting for the largest proportion in the study total area, mainly including high geological hazards, permanent basic farmland, rivers and water area, etc.

### 3.2. Analysis of Township Construction Land Carrying Capacity

#### 3.2.1. Analysis of Township Construction Land Carrying Capacity

The critical value of construction land in Xifeng District (utmost development intensity) is 52.62%, and the critical value of construction land of each township is 3.78%~13.15%, which reflects the significant difference in the value of each township (Table 3). Among them, the townships with the critical value of carrying less than 6% involve Shishe and Xiansheng Town. Due to the influence of the landforms in the Loess Plateau hilly and gully region, the proportion of areas with high water and soil erosion and geological hazards are relatively large, so that the development intensity of township construction land is limited and the carrying capacity of the land is relatively weak. The townships with the critical values of carrying capacity between 6% and 10% include Houguanzhai, Pengyuan, Wenquan, and Xiaojin. Although the townships are affected by natural disasters, based on their location conditions, the central and key villages in the townships are capable of attracting population agglomeration from the township or surrounding townships, which promotes the development and utilization of land, compared with Shishe and Xiansheng, whose development intensity is higher. More than 10% of the townships are only Dongzhi, which mainly relies on the radiation of the central city and the relatively good foundation for development of the secondary and tertiary industries with the highest limit development intensity. The spatial distribution pattern of the carrying capacity of construction land in Xifeng is shown in Figure 4.

#### 3.2.2. Analysis of Township Construction Land Development Potential

Owing to the different conditions of townships in the study area, such as the geographical location, development foundation, and dominant function, the development intensity of the construction land was between 0.70% and 2.84%. The value of each township was significantly different (Table 3). Among them, Dongzhi, Houguanzhai, Pengyuan, and Wenquan, which are adjacent to the central urban area, developed between 1.52% and 2.84%, due to their location advantages and economic conditions. Xiaojin, as the urban area of Qingyang, leads to Xi’an, and it is an important node in large and medium-sized cities, such as Lanzhou, Pingliang, and other large and medium-sized cities in Qingyang city, and important regional distribution center for people, logistics, information, and commodities. Its development intensity is 1.69%. The development intensities of Shishe and Xiansheng, far away from the city area, are 0.94% and 0.70%, respectively.

According to the ultimate development intensity and the developed intensity of the township’s construction land in Xifeng District, the remaining development intensity of the land is 41.42%. This can be seen in Table 3. The development potential of each township construction land in Xifeng District is between 3.08% and 10.32%, which shows remarkable difference. Among them, although most of the administrative areas of Dongzhi, Wenquan and Houguanzhai are located on the flat terrain, on account of the constraints of geological disasters and permanent basic farmland delimitation, the development intensity is relatively small, and the remaining development intensity is relatively large. The development intensities of the remaining ones are 10.32%, 5.58%, and 5.98%, respectively. Based on the advantages of convenient external transportation and logistics, the development intensities of Pengyuan and Xiaojin town are 1.89% and 1.59%, respectively, although they are also limited by the geomorphology and protection reserve. The remaining development intensities are also relatively large at 6.58% and 6.47%, respectively. For Shishe and Xiansheng, due to the common constraints of topography, spatial location, and transportation conditions, their ultimate development intensity and remaining development intensity are relatively small, and the remaining development intensities are 3.42% and 3.08%, respectively. Therefore, in the allocation of construction land indicators, priority should be given to townships with greater remaining development intensity and better suitability in the future, such as Dongzhi and Wenquan, etc. It is convenient to achieve intensive and efficient use of construction land and promote regional sustainable development.

### 3.3. Spatial Pattern Matching Mnalysis of Township Construction Land

The spatial pattern matching degree reflects the development status of construction land and the degree of spatial coupling that could carry construction land. The current construction land in the study areas is 143.86 km^2^. Among them, the most suitable carrying area is 45.67 km^2^, the more suitable carrying area is 31.76 km^2^, the generally suitable carrying area is 12.72 km^2^, the less carrying area is 36.12 km^2^, and the unsuitable carrying area is 17.59 km^2^ (Figure 5). The proportion of current construction land in each township ranges from 4.78% and 19.52%, with significant differences. The smallest proportion of the current construction land is in Xiansheng, and the largest is Dongzhi. The current construction land area accounted for less than 10% in Xiansheng and Shishe. In Houguanzhai, Wenquan, Pingyuan, and Xiaojin, the values are between 10% and 15%, and it is more than 15% in Dongzhi Town.

According to the spatial superposition results of the current construction land and the suitability comprehensive evaluation result, the spatial pattern matching degree of the township construction land in Xifeng District can be analyzed (Figure 6). By calculating the ratio of the current construction land of each township located in the suitable construction area to the total area of the current construction land in the region, we obtain the spatial pattern matching degree of each township is between 71.44% and 85.55%, which indicates that the spatial pattern matching degree of each township is moderate and the difference is small. Among them, the spatial pattern matching degrees of Dongzhi, Houguanzhai, and Wenquan are 85.55%, 84.49%, and 84.44%, respectively, which mainly depend on their superior geographical location and topographical conditions. The spatial pattern matching degree of Pengyuan and Xiaojin are 75.76% and 83.89%, respectively, because a logistics park of a certain scale, which promotes the land development intensity that has been built and developed external transportation conditions. The spatial pattern matching degree of Shishe and Xiansheng are 71.44% and 72.04%, respectively; their lands are restricted, and the matching degree is small because of ecological protection and topographic conditions. The overall spatial pattern matching degree of the study area is 81.32%, which indicates that the overall carrying suitability of the current township construction land in Xifeng District is collectivity good.

In addition, based on the correlation analysis results between the critical value of the construction land and the space pattern matching degree in Xifeng District, it is known that there is a positive correlation between the critical value of construction land and the degree of the space pattern matching. It can be seen that the larger the critical value, the higher the space matching degree, which reveals the matching relationship between the current development and the rational development.

## 4. The Discussion and Conclusions

### 4.1. Discussion

According to the research results, there is still much room to improve the land carrying capacity of Xifeng. Due to the uneven regional development and different geographical environment, each township has different land carrying capacity and spatial characteristics, which can effectively express the matching degree of land with other resource elements and the spatial disposition status of production and living activities and can provide more effective decision-making support for spatial planning and control. Besides, different measures should be taken and adjusted, in order to adapt to the local conditions in different towns. For instance, the control system on the usage of land should be strictly implemented to the areas within the boundaries of urban development and industrial parks, the transformation from agricultural land to industrial land should be restricted, and the potential of stock construction land is the main direction of township land use [78]. For areas with serious water and soil loss and high risk of geological disasters, the policy of returning farmland to forest should be actively implemented and afforestation should be vigorously developed, excluding water and soil conservation and control projects [79]. For some regions with backward traffic conditions, the mode of economic development and land use should be appropriately transformed, and traditional agriculture should be promoted and transformed into ecotourism and high value-added ecological agriculture could be developed.

Compared with previous studies, this study was based on regional reality, took Xifeng as the research area, townships as the research scale, and construction suitability evaluation as the entry point to explore the matching of the construction land carrying capacity and spatial pattern of the township units in the hilly and gully regions of the Loess Plateau. The research results attempt to provide some references for the optimization of the spatial layout of the township units and the intensive and economical use of land resources in the Loess Plateau hilly and gully regions. Meanwhile, research on land carrying capacity can provide ideas for the construction of international ecological security pattern and sustainable development. However, it should be pointed out that there are still some deficiencies in the research. First of all, this study only selected one county in the Loess Plateau hilly and gully region for empirical analysis, with one time section. More sample units and longer times of carrying capacity of township construction land in the Loess Plateau hilly and gully region need further exploring, as well as the spatio-temporal differentiation of spatial pattern matching degree. In addition, it is worth thinking about the next step to explore the evolution characteristics and optimization strategies of the carrying capacity of township construction land in the hilly and gully regions of the Loess Plateau under different scenarios in the future.

### 4.2. Conclusions

Based on the suitability evaluation of township construction land, this study measured the current carrying capacity level of township construction land in Xifeng, through the viewpoints put forward by predecessors, and used spatial analysis method to explore the spatial differentiation characteristics and spatial pattern matching of township construction land carrying capacity. The main conclusions are as follows: ① The spatial suitability and concentration of construction land of townships in Xifeng are quite different, and the overall performance is as follow: adjacent city areas > city and urban radiation areas > remote mountain areas. ② The critical value of overall carrying capacity of township construction land in Xifeng is 52.62%, the carrying capacity of each township is between 3.78% and 13.15%, and the value of carrying capacity varies significantly. The suitable carrying area of construction land of each township is mainly concentrated on the flat tableland and both sides of the traffic trunk line. ③ The overall spatial matching degree of towns in Xifeng is 81.32%, which indicates that the current land development is in good condition. The spatial form of construction land in townships is mainly in sheet, strip, and dot, showing a distribution pattern of “small agglomeration and large dispersion”.

### 4.3. Suggestion

Based on the discussion on the carrying capacity of township construction land and the matching degree of spatial pattern in Xifeng District, this paper tries to put forward the countermeasures and suggestions for improving the carrying capacity of township construction land from the following aspects. ① Accelerate the upgrading of industrial structure. Ensuring stable economic growth is an important basis for improving land carrying capacity. For hilly and gully areas on the Loess Plateau with lagging economic development, more attention should be paid to the development of township leading industries in the future, for example, actively cultivating high-tech industries with development potential, strengthening financial investment in scientific and technological development, and improving the level of technology to attract foreign investment. ② Improve the social security system. On the one hand, we should strengthen the road construction in remote mountainous areas, vigorously develop public transport, and improve the efficiency of residents; what is more, we should promote the equalization of urban and rural public service capacity and optimize the distribution system of urban and rural health care, education, and culture. ③ Strengthen ecological environment protection. Ecological environment is the key factor affecting the sustainability and continuity of land carrying capacity. For hilly and gully areas of the Loess Plateau, at first, we should improve the ability to prevent natural disasters and minimize the economic losses caused by natural disasters; furthermore, we should vigorously develop green and ecological agriculture, prevent water and soil loss, strengthen the control of production and domestic pollution sources, actively promote the delineation of town ecological space, effectively promote the ecological environment capacity of villages and towns, and then develop the land carrying capacity of villages and towns.

## Figures and Tables

**Figure 1 ijerph-19-16316-f001:**
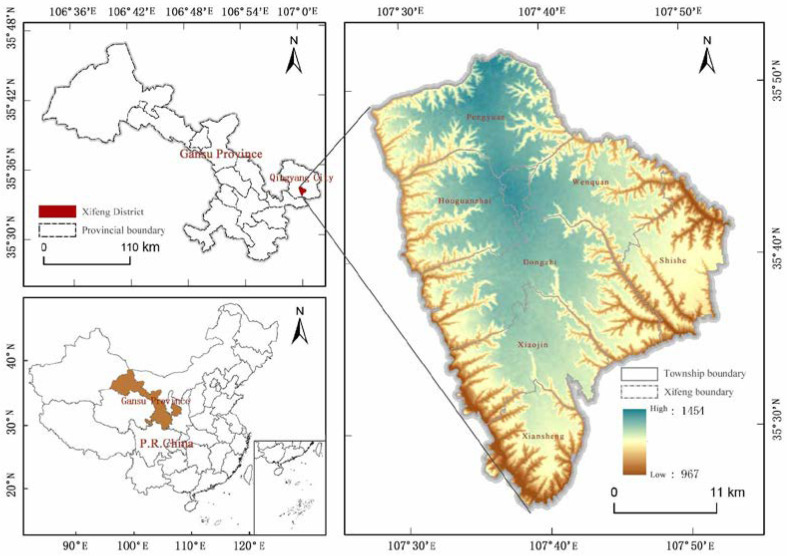
Study area.

**Figure 2 ijerph-19-16316-f002:**
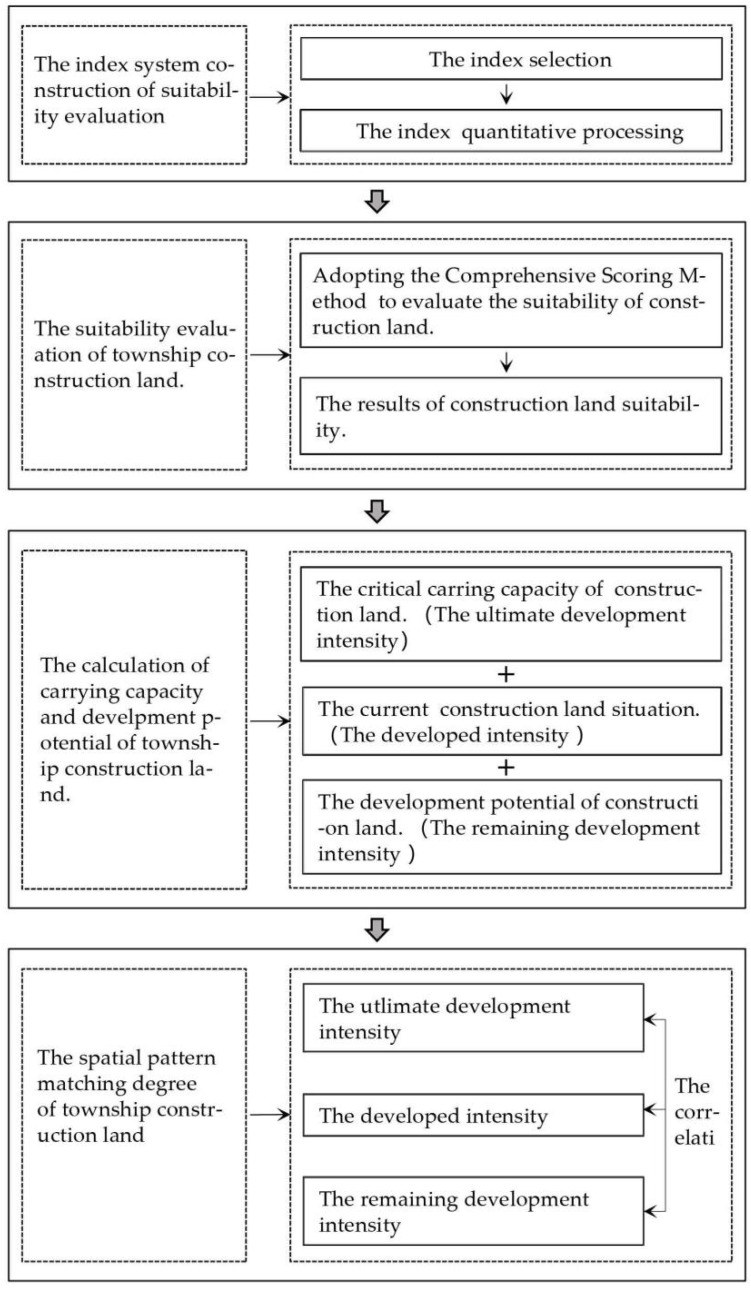
The process of the research of township construction land carrying capacity and spatial matching in Xifeng.

**Figure 3 ijerph-19-16316-f003:**
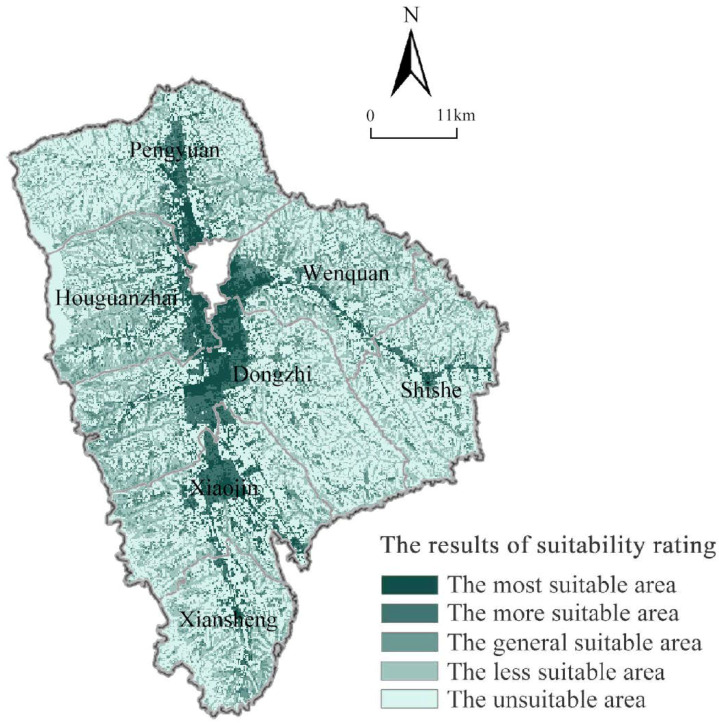
The suitability grade of township construction land in Xifeng.

**Figure 4 ijerph-19-16316-f004:**
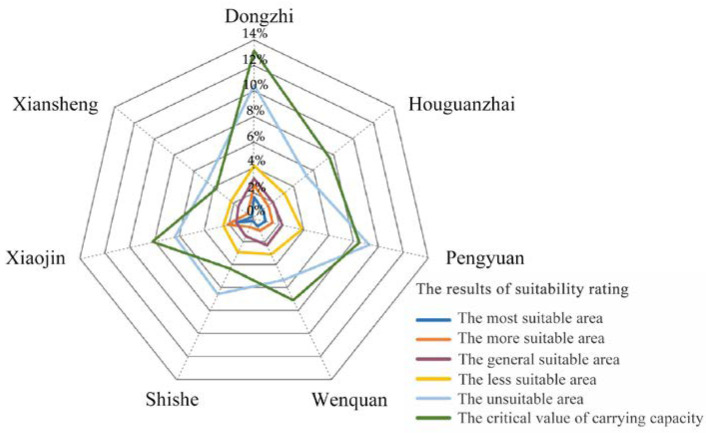
The statistics of suitability area and carrying capacity of township construction land in Xifeng.

**Figure 5 ijerph-19-16316-f005:**
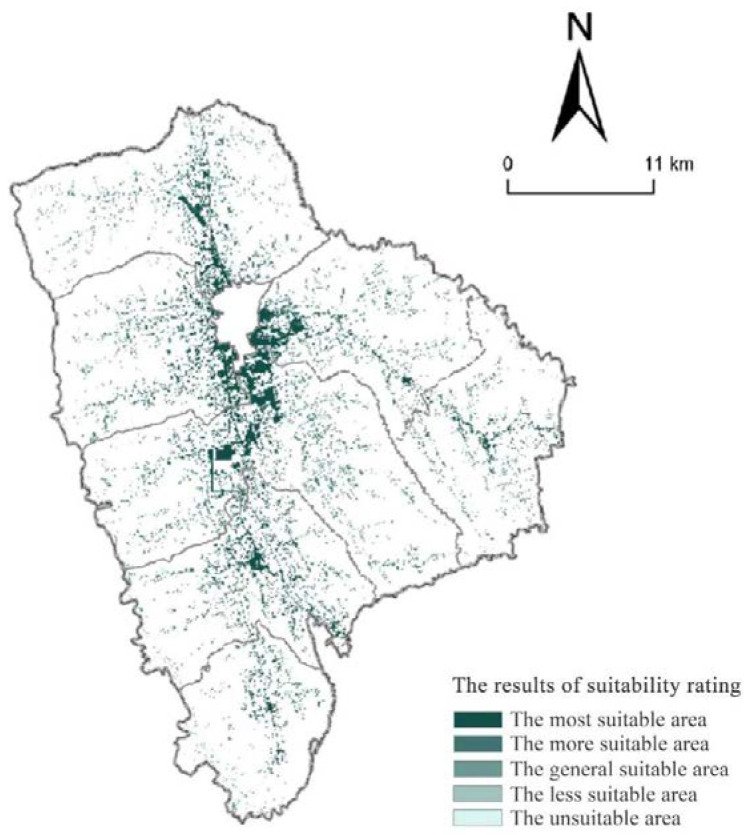
The suitability grade of the current construction land of townships in Xifeng.

**Figure 6 ijerph-19-16316-f006:**
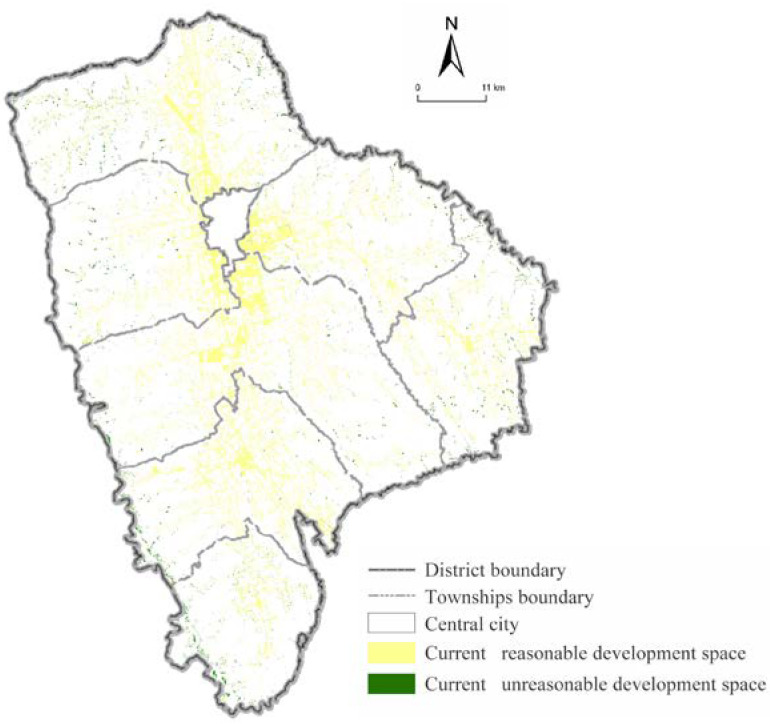
The current development status of construction land in Xifeng.

**Table 1 ijerph-19-16316-t001:** Evaluation index system for the appropriateness of township construction land.

Indicator Type	Middle Layer	Index Factor	Factor Grading	Assignment	Weights
The strongly binding indicators	The ecological protection	The ecological protection red line	The ecological protection red line	0	
			Other	1	
		The permanent basic farmland	The permanent basic farmland	0	
			Other	1	
		Water areas	Water areas	0	
			Other	1	
The less strong binding indicators	The natural environment	Slope	>25°	1	0.1193
			15°–25°	3	
			8°–15°	5	
			3°–8°	7	
			0–3°	10	
		Elevation	>1400 m	1	0.0327
			1250 m–1400 m	2	
			1100 m–1250 m	5	
			1000 m–1100 m	7	
			0–1000 m	10	
		Geological disasters	The high zone of geological disasters	2	0.3365
			The middle-zone of geological disasters	6	
			The low zone of geological disasters	8	
			No geological hazard zone	10	
		The current land use	Gardens, woodlands, wetlands	3	0.2224
			Other land	5	
			Grassland, unused	7	
			Construction land	10	
	The social economy	The distance from national highway	>5000 m	1	0.0845
			3500 m–5000 m	3	
			2000 m–3500 m	5	
			1000 m–2000 m	7	
			0–1000 m	10	
		The distance from provincial roads	>3000 m	1	0.0742
			2000 m–3000 m	3	
			1000 m–2000 m	5	
			500 m–1000 m	7	
			0–500 m	10	
		The distance from county roads	>1500 m	1	0.1304
			1000 m–1500 m	3	
			500 m–1000 m	5	
			300 m–500 m	7	
			0–300 m	10	

**Table 2 ijerph-19-16316-t002:** Xifeng’s township construction land suitability of hierarchies and area ratio.

The Rating of Suitability Evaluation	The Divided Standard	Areas (km^2^)	Proportion (%)
The most suitable area	7.63–9.77	58.15	5.88
The more suitable area	6.14–7.63	100.47	10.15
The general suitable area	4.83–6.14	140.33	14.18
The less suitable area	3.58–4.83	215.87	21.82
The unsuitable area	2.18–3.58	474.69	47.97

**Table 3 ijerph-19-16316-t003:** The development intensity of townships construction land in Xifeng.

Administrative Area	The Ultimate Development Intensity (%)	The Developed Intensity (%)	The Remaining Development Intensity (%)
Dongzhi	10.32	2.84	7.48
Houguanzhai	5.98	1.63	4.35
Pengyuan	6.58	1.89	4.69
Wenquan	5.58	1.52	4.06
Shishe	3.42	0.94	2.48
Xiaojin	6.47	1.69	4.78
Xiansheng	3.08	0.70	2.38

## Data Availability

Not applicable.

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
