# Peer review of "Study of Township Construction Land Carrying Capacity and Spatial Pattern Matching in Loess Plateau Hilly and Gully Region: A Case of Xifeng in China"

_ijerph, 2022, doi:10.3390/ijerph192316316_

Round 1

Reviewer 1 Report

Dear Authors

First of all, I have to express that authors have deserved an appreciation and a thank for writing this article. I am glad to read it, actually my intention for all reviewed articles by me to let the article to be published in case providing efforts. I have done the same behaver to this article. But still there are some disadvantageous in it. I have tried to explain them in my review. Would you please, take that explanations into your account and force yourselves to overcome each of them in a good manner. Please do not forget, to be published an article is not hard job, you are of course able to do it either this journal or another. But it is difficult to the other authors to convince them to refer your article, and, it depends only on your article quality. My duty is to make a provision/decision to turning that article into much better version comparing it to previous version. If you are inclined to revise them to be published it ASAP, you can do that according to my review report. All in all, you are able to read my evaluation. Please revise all of them to persuade me and/or the Editor.

Regards.

Comments

Abtsract

I have expected that the information given in the abstract part of the article should have reflected a small summary of the study. However, in its current form, the summary section, the importance of the study in the world literature, the definition of the problem, the purpose of the study, the findings and the most important result should be rewritten with an important suggestion. For these reasons, I have to recommend the authors to re-organize and / or re-write the summary of the study.

Line 35

In this section, would you please talk about the literature studies on land and land carrying capacity. Then, would you please include previous scientific studies and sample applications on these subjects in your work. Next, would you please relate all of this to your home country and work area.

Line 48

Do you think this concept, which you describe as land carrying capacity, should be considered as one of the data layers that are the base of the plan before spatial planning? Or should the land carrying capacity be calculated and included in the plans after the spatial plan is made? Please provide detailed information on this subject at this stage. Discuss the land bearing capacity and its relation to spatial plans.

Line 68

For this section, consider the site selection analysis methods in the scientific research articles that I have shared in detail below. Again, please explain the practices in your own country by giving information from the studies conducted in this literature. Are there any land carrying capacity calculation methods that have been discussed with the methods in the studies I suggested in this section?, Please inform us about that topics. (Evaluation of offshore wind power plant sustainability: A case study of Sinop/Gerze, Turkey"; DOI: 10.1504/IJGW.2021.114342)

Line 82

I should suggest that you design and write this section under two subheadings. After clearly describing the problem in the first sub-title, state the purpose of your work in the second sub-title.

Line 104

Please convert Figure 1 into a good format suitable for a scientific research article. For this, rearrange the figure's find-location map. For this, it will be sufficient to look at the map in the article I gave as an example above (line 68).

Line 114

Consider also each of the land types described here in terms of whether or not they can be privately owned. In the spatial planning stage in China, how are such lands evaluated according to the relevant legislation? While making these explanations, you can benefit from the types of immovables, whether these types are subject to ownership, and immovable classes in the below article I have given. It will be very helpful for the clarity of your article if you present the current situation in China.

Line 116

You can give the name of the program you use (ArcGIS…) at the end of the article rather than in the article. However, I don't think you need to give the program name either. As it is known, there are many similar software and all of them are sufficient to carry out the same operations. If you want to give this name, the discretion is yours.

Line 136

I think that the figure 2 should be rearranged since the figure 2 has not been prepared meticulously.

Line 146

Are the regions you mentioned in this paragraph among the land classes defined as protected areas within the scope of UNESCO? Please explain this in accordance with the literature. Regarding protected areas, you can use the article I have given below. (WOS, DOI: 10.1016/j.landusepol.2022.106357, Protected area geographical management model from design to implementation for specially protected environment area )

Line 182

What is stated in this section is actually similar to the articles in which the appropriate site selection for the construction of facilities such as a solar farm or a wind farm is made. In this type of articles, the criteria required for the field and the weights of the criteria are determined. According to these criteria, the most optimum area is determined on the basis of comparison in certain regions. Then, the production capacity of the facility to be built in this area is calculated, and a cost-benefit analysis is made. Ultimately, the work done in this study is the same. This work is perceived as the preparation of a optimum site selection map. If what I have said is true, you have mentioned the literature examples related to appropriate site selection in this section. The article I recommend above (Line 68) and many other articles that you can access from WOS will help you with this.

Line 191, 209, 223, 237

This explanation should be considered separately in the formulas on lines 191, 201, 223 and 237. Did you discover the formulas in this section yourself? Or are they formulas that have been used before in other people's work? And have you applied these formulas to your own work? Please explain this situation. Also, explain the formula with numerical data in a very small working area below the formula. Prepare a simple example with a solution. I should also suggest that you do a little calculation for this. The formula is not very clear.

Line 298

Why was Figure 4 prepared in this format? Is it supposed to be like this as a custom display type? I could not understand. Does this shape have a special meaning? Can the data here be explained in more detail in a different way? In conclusion, what is this figure trying to explain? You should explain it more simply and carefully rearranging it.

Line 373

Although there is a discussion statement in the title and subtitle of this section, it cannot be said that there is a discussion in the section. Because the previous similar or different studies in the literature have not been compared and evaluated. This section should be given under the heading of results, it needs to be reconsidered in this section according to the corrections mentioned above.

At the end

You are expected to suggest something in the last paragraph of the work. What do you suggest at the end of this study? You should write your suggestions for new studies and studies to be carried out by other scientists/researchers by addressing the application areas and application forms of what you have obtained as a result of the study with three or five sentences.

Author Response

Please see the attchment.

Reviewer 2 Report

The reviewed article is very interesting. I would like to point out several elements that should help the authors improve the study. 

The presented article, in the reviewer's opinion, needs to be cleaned up both the abstract and the main body of the article, which should make it clearer.   

Please indicate the originality of the presented research. How can the results of the analysis be used? Do the results of the analysis provide advances in current knowledge, and to what extent? Can the results of the analysis be used in other countries, and to what extent? 

Check the formatting of the literature. 

Abstract. Reformat the abstract to summarize the article well.

Needs correction, requires new writing in the reviewer's opinion. It should be structured. Its structure should include 1) the research problem (the topic of the work mentioned in a general way), 2) the purpose of the work, 3) the method and research area, 4) a general description of the results of the research (general conclusions without describing them, what is their contribution to business practice)

No indication of the method and purpose of the study. 

Introduction. In the reviewer's opinion, it has the character of a literature review. The introduction should clearly illustrate what we know in terms of the topic addressed, and what we do not know - what is new in the study (why the topic should be addressed).

The reviewer proposes to give more indication of the purpose, research questions (which will facilitate its evaluation). Expand the introduction to clearly state the research problem in order to adequately inform the reader.

About the TOPSIS method, it seems that it should be included in the section describing the methodology. Maybe it would be possible to show it in a mathematical way (formulas). About the method they wrote: 

https://doi.org/10.3390/su142114007

https://doi.org/10.3390/plants11212827

https://doi.org/10.3390/su142013690

https://doi.org/10.3390/w14203262

https://doi.org/10.3390/su142113824

The introduction should be more divided into an introduction and a literature review. This should better show the purpose of the study and its relevance to the economy.

If the literature is separated, it seems reasonable to indicate in it also the elements of quality of life, green economy, green infrastructure, sustainable development, which should also be, according to the reviewer, part of the indication of the selection of variables for analysis. In the area indicated, they wrote 

https://doi.org/10.3390/ijerph19159185

https://doi.org/10.3390/land11101792

https://doi.org/10.3390/land11101765

https://doi.org/10.3390/app121910189

https://doi.org/10.3390/ijgi11100513

https://doi.org/10.3390/su141912618

https://doi.org/10.3390/su142113977

https://doi.org/10.3390/su142113971

https://doi.org/10.3390/su142113971

Literature Review. It would be necessary to indicate the theoretical framework of the research process carried out.  

The article should address issues of real-world relevance in a coherent and compelling manner. The theoretical framework emerging from the literature review could explore questions and points of emphasis.

 Study Area and Data. Train of thought and Methodology. The two parts that describe method and material need to be organized. The explanation of research procedures with a corresponding explanation of methods.

The analysis of results. Requires ordering. 

The conclusion and discussion

Discussion. There should be a reference to the research results of other authors. What the reviewer does not notice in the reviewed material. At the end of the Discussion section, please add the strengths and limitations of this study 

The discussion is in addition to the literature review found in the introduction 

Conclusion. In addition to general conclusions, references to the results obtained, an indication of the problems the authors had during the analysis, or who can use the research, In the Conclusion section, Please indicate the originality of the research presented.  Do the results of the analysis provide an advance in current knowledge, and to what extent? Can the research be related to the international literature, and to what extent?

Reviewer 3 Report

First of all, this paper was very difficult for me to review because of the poor language. Many parts and sentences were impossible to understand, and in many cases, it required a lot of effort to figure out what the authors intended to say. Just for this reason, the paper should be rejected.

Nevertheless, there are also other arguments for the rejection.

The paper is mostly descriptive and methodological and thus lacks scientific soundness and significance. The literature review (limited almost exclusively to the Chinese papers) is extremely superficial, limited to citations but without any review of the previous results. The research does not have any theoretical background as well as a hypothesis, while the paper’s (research) aim is formulated as purely descriptive – to calculate the carrying capacity and development potential. The authors focused entirely on the method without any reflection on what will be the scientific value and meaning of the obtained results. It is even more evident when we look at the Discussion section which is extremely modest and does not meet any standard criteria for this section in a scientific article.

Therefore, the paper does not fulfil the basic requirements of a scientific paper and should not be published in a scientific journal.

There are also numerous other, minor shortcomings, to mention only a few:

-          The map of the study area does not provide sufficient information about where exactly the area is located (it should be presented on the country map); the legend on the right-hand side map lacks labels

-          The authors use the strange title of one of the sections e.g. Train of thought

-          The authors represent a very mechanistic and mathematical approach to the idea of spatial planning. I understand that such an approach might be useful but should be certainly presented in a critical context and perspective i.e. explaining and analysing its limitations and conditions

-          One of the conclusions of the study is that the current township construction land in Xifeng District is good, even though it was not based on such scientific analysis as presented in the paper (actually, on the map not a single green area is visible); so what is the point in conducting such analysis?

Round 2

Reviewer 1 Report

thank you for all revisions

Author Response

Dear Reviewer:

Thank you for your approval of the comment response. I wish you good health,everything goes well with your work!

Reviewer 3 Report

The article was improved although I still think that the theoretical background and reference to previous research are not sufficient. I see that the authors worked on that and I appreciate their effort however, the discussion section still lacks reference to (and discussion with) the previous research results.
